# Applications of Au_25_ Nanoclusters in Photon-Based Cancer Therapies

**DOI:** 10.3390/nano15010039

**Published:** 2024-12-29

**Authors:** Zoey A. Lockwood, Michael R. Jirousek, James P. Basilion, Clemens Burda

**Affiliations:** 1Department of Chemistry, College of Arts and Sciences, Case Western Reserve University, Cleveland, OH 44106, USA; 2Department of Radiology, School of Medicine, Case Western Reserve University, Cleveland, OH 44106, USA

**Keywords:** gold nanoclusters, targeted cancer therapy, radiotherapy, photodynamic therapy, photothermal therapy, PDT, PTT, AuNC

## Abstract

Atomically precise gold nanoclusters (AuNCs) exhibit unique physical and optical properties, making them highly promising for targeted cancer therapy. Their small size enhances cellular uptake, facilitates rapid distribution to tumor tissues, and minimizes accumulation in non-target organs compared to larger gold nanoparticles. AuNCs, particularly Au_25_, show significant potential in phototherapy, including photothermal (PTT), photodynamic (PDT), and radiation therapies. These therapies benefit with minimal damage to surrounding healthy tissue. AuNCs also demonstrate excellent stability and biocompatibility, crucial for their effective use in clinical applications. Recent advances in the synthesis and functionalization of AuNCs have further improved their therapeutic efficacy, making them versatile agents for enhancing cancer treatment outcomes. Ongoing research aims to better understand their pharmacokinetics, biodistribution, and long-term safety, paving the way for their broader application in advanced cancer therapies.

## 1. Introduction

The range of sizes exhibited by gold nanoparticles often leads to the emergence of unique physical, chemical, and optical properties, alongside different interactions with biological systems [1,2,3,4,5,6,7]. Over the past decade, our group has studied the fascinating area of gold nanoparticles (AuNPs) in the context of cancer therapy [8,9,10,11,12,13,14]. The group has demonstrated the use of targeted AuNPs for precise drug delivery and enhanced photon-therapy in various cancer models, including brain and prostate cancer. This work showcases the potential of AuNP-based approaches to improve the effectiveness and precision of cancer diagnosis and treatments. In recent research from 2019, our group focused on the size effect of targeted AuNPs. Luo et al. found that the size of targeted AuNPs significantly affects cellular uptake and in vivo bio-kinetics. Smaller Au nanoconjugate systems (2 nm) exhibit a prolonged circulation period in vivo, facilitating aggregation in tumors. Silver staining analysis demonstrated that 2 nm AuNPs have a more uniform dispersion and deeper penetration into cells and tumors. Additionally, smaller AuNPs result in lower liver uptake, potentially reducing off-target elemental toxicity of gold [15]. These findings highlight the advantages of smaller AuNPs, indicating that exploring gold nanoclusters (AuNCs) as a potential candidate for a targeted cancer therapeutic is essential. Additionally, no two NP batches are precisely the same, which presents a major obstacle for reliably controlling their properties [16].

To overcome these issues, we have turned our attention to atomically precise AuNCs. AuNCs are ultrasmall clusters consisting of less than 100 gold atoms, with a diameter of less than 2 nm, and they are being explored as stable and controllable alternatives to larger nanoparticles [17]. One of the most popular AuNCs is Au_25_, a cluster composed of 25 gold atoms that are typically functionalized with 18 ligands on the surface [18,19,20,21,22]. The properties of Au_25_NCs, along with those of other nanoparticles of varying sizes, are summarized in Table 1, illustrating the comparative advantages of these AuNCs in terms of synthesis, reproducibility, toxicity, cellular uptake, and therapeutic efficacy. The data in Table 1 indicate that nanoparticles across all size ranges can be synthesized with relative ease [19,23,24]. However, the ultrasmall size and atomic precision of Au_25_NCs offer additional advantages in terms of reproducibility [25,26]. The ability to consistently produce Au_25_NCs with well-defined sizes and compositions provides a significant edge in clinical applications, where uniformity is paramount for reliable results. Larger nanoparticles, while also easy to synthesize, often display more variability in size, which can impact their performance [23,27,28,29]. One of the most notable advantages of Au_25_NCs is their lower toxicity [30,31]. Some of the key advantages of Au_25_NCs include their biocompatibility and efficient elimination from the body [26,30,31]. Their ultrasmall size enables renal clearance, minimizing long-term retention and thereby reducing the risk of chronic toxicity, a common concern associated with larger nanoparticles [27,32,33]. The reduced toxicity of Au_25_NCs is particularly critical for clinical translation, highlighting their potential as safer alternatives in nanomedicine. In addition to their safety profile, Au_25_NCs demonstrate enhanced cellular uptake compared to larger particles [34,35]. Their ultrasmall size facilitates rapid distribution and in some cases endocytosis, enabling them to penetrate cells and deliver therapeutic or imaging agents effectively [35,36]. Moreover, Au_25_NCs show superior tumor uptake and retention, likely due to their ability to penetrate dense tumor tissues more effectively than larger particles [15,37]. This enhanced uptake at the tumor site is critical for improving the therapeutic efficacy in cancer treatment. Their enhanced tumor uptake increases their efficacy upon radiation when compared to larger nanoparticles, making them more effective for photon-based therapies [38,39]. Finally, the elimination of Au_25_NCs further sets them apart. Their renal clearance ensures minimal long-term retention in the body, reducing the likelihood of chronic toxicity and addressing a common concern associated with larger nanoparticles [26,31,40]. Overall, Au_25_NCs outperform larger AuNPs due to their superior reproducibility, enhanced cellular and tumor uptake, and rapid clearance, which minimizes systemic toxicity. These properties make them promising candidates for safer and more effective applications in cancer treatment and photon-based therapies.

Over the past five years, Au_25_NCs have undergone investigation for their promising potential in cancer treatment. These ultrasmall clusters not only exhibit superior biocompatibility and stability but also offer a unique platform for photon-based therapies. Their optical and electronic properties make them particularly suited for innovative therapeutic approaches such as photothermal therapy (PTT), which utilizes heat generation upon light absorption, and photodynamic therapy (PDT), which leverages the generation of reactive oxygen species to destroy tumor cells [47]. Furthermore, ultrasmall AuNCs, with their strong absorption and ability to generate secondary electrons under irradiation, serve as effective radiosensitizers by enhancing tumor damage during radiotherapy while ensuring efficient clearance to minimize damage to healthy cells [38]. These therapeutic approaches, driven by the unique properties of Au_25_NCs, represent a potential advancement in the development of precision cancer treatments. However, the realization of their full potential hinges on meticulous synthesis and characterization processes, which ensure their stability, reproducibility, and functional optimization for biomedical applications. The following section delves into the synthesis and characterization of AuNCs, highlighting key methods and their impact on clinical translatability.

## 2. Synthesis and Characterization of AuNCs

Significant advancements have been made in the past five years to achieve controlled and high-yield synthesis of size-tailored AuNCs. Fluorescent AuNCs have been successfully synthesized through one-step reactions employing various reducing agents. Two commonly used peptides in AuNC synthesis are glutathione (GSH) and bovine albumin serum (BSA) [30,48,49,50,51]. Additionally, Bhamore et al. utilize pepsin as a potential reductant and stabilizing agent shown in Figure 1 [52]. Characterization involved ultraviolet–visible (UV-Vis) and fluorescence spectroscopy, high-resolution transmission electron spectroscopy (HR-TEM), dynamic light scattering (DLS), zeta potential measurements, and Fourier-transform infrared spectroscopy (FT-IR). The absorbance spectrum of P-AuNCs displayed a distinct peak at 328 nm, different from the absorption spectra of pepsin and HAuCl_4_. The lack of localized surface plasmon resonance (LSPR) features in the visible spectrum, commonly observed in larger AuNPs, highlights pepsin’s effectiveness in reducing and stabilizing AuNCs. Further characterization found that P-AuNCs exhibited strong emission at 655 nm. The stability of P-AuNCs over 45–60 days was confirmed, supported by HR-TEM, DLS, and zeta potential measurements. FT-IR indicated Au^3+^ interaction, leading to red-emitting P-AuNCs.

Li et al. focus the synthesis and characterization of alkynyl-protected Au_25_NCs, using Me_2_AuCl to introduce chloride during the synthesis. Silica gel column chromatography effectively isolated Au_25_, as well as two additional clusters, Au_67_ and Au_106_, revealing the potential of chloride as a stabilizing ligand [53]. Dou et al. explored a modified two-phase Brust–Schiffrin (B-S) approach for synthesizing Au_25_NCs shown in Figure 2. In this approach, tetraoctylammonium bromide (TOAB) facilitates the transfer of both Au(III) precursors and NaBH_4_ from aqueous to organic phases. This technique introduces hydrophobic selenolate ligands for Au(III) transfer and precisely controls TOAB addition for NaBH_4_ transfer. The absence of TOAB micelles enables control over reduction kinetics, resulting in size-tunable AuNCs with high yields and short synthesis times characterized by electrospray ionization mass spectrometry (ESI-MS) and UV-Vis spectroscopy [54].

Step 1 transfers AuCl_4_^−^ from the aqueous to organic phase through coordination with thiol-based ligands (R-SH), forming Au(I)-ligand complex. Step 2 involves reductive BH_4_^−^ transfer aided by TOA^+^ through electrostatic interaction (TOA^+^-BH_4_^−^). The reaction shows the reduction of the gold–ligand complex, leading to the formation of ligand-protected AuNCs. AuNCs can also be synthesized using a CO-reduction method. For example, Yao et al. employed the CO-reduction method, combining aqueous solutions of p-MBA and HAuCl_4_, adjusting the pH to 10.5 with NaOH, and introducing CO into the reaction mixture to synthesize [Au_25_(p-MBA)_18_], characterized by UV-Vis spectroscopy and ESI-MS- [55]. Similarly, Cao et al. employed the CO-reduction method by combining aqueous solutions of MHA and HAuCl_4_, adjusting the pH to 12.0 with NaOH, and introducing CO gas to yield [Au_25_(SR)_18_]^−^. However, they also demonstrated ligand engineering can be achieved through a reversible process that enables the addition and elimination of a single thiolate ligand on AuNCs. Through oxidative etching of [Au_25_(SR)_18_]^−^ NCs, an additional thiolate ligand leads to the creation of a new species, [Au_25_(SR)_18_]^0^, and this modification can be reversed via CO reduction again. These different species were characterized through UV-Vis spectroscopy and ESI-MS [56]. Hossain et al. focus their study on ligand exchange reactions for introducing new properties to Au_25_NCs. They explore exchange reactions involving chalcogenates, such as PhSeH or (PhTe)_2_ as ligands, revealing preferential production of products with substituted ligands close to the gold core [57]. Chen et al. investigate influence of temperature on the synthesis of Au_25_(SR)_18_NCs. They utilize a heated synthesis protocol to efficiently produce thermodynamically stable Au_25_(SR)_18_NCs with high yield (>95% on gold atom basis) and fast kinetics. This approach effectively balances and accelerates the reduction-growth and size-focusing reactions, resulting in optimal results at 40 °C. This leads to the efficient synthesis of Au_25_(SR)_18_NCs [58].

In the past five years, significant progress has been made in the controlled, high yield synthesis of size-tailored AuNCs. Various methods, including one-step reactions with reducing agents and peptides like GSH, have produced AuNCs. Notable approaches involve alkynyl and selenolate ligands, CO reduction, and reversible ligand engineering, with increased temperature optimizing yield.

### 2.1. Synthesis and Characterization of Targeted AuNCs for Cancer Therapy

AuNCs have gained significant attention for cancer therapy due to their ability to selectively accumulate in tumor tissues through surface functionalization. By engineering their surfaces with targeting ligands, peptides, or biomolecules, AuNCs can achieve precise tumor targeting while minimizing off-target effects and reducing systemic toxicity. Additionally, the synthesis of these AuNCs enables control over important properties such as size, stability, and optical characteristics, which are all important properties to consider for cancer therapy.

Kong et al. utilize a protein-templated synthesis using RNase-A by forming an Au^3+^-protein complex through the reduction of AuCl_3_ in the presence of RNase-A, followed by pH adjustment and incubation at 37 °C. Covalent conjugation with Vitamin B12 (VB12) using EDC/NHS coupling yields VB12-R-AuNCs, which target receptors overexpressed in cancer cells, promoting efficient internalization and offering potential for orally delivered tumor imaging. Successful synthesis and structural integrity were confirmed through UV-Vis spectroscopy and fluorescence spectroscopy, which showed characteristic absorption and emission peaks. MALDI-TOF-MS (Matrix-Assisted Laser Desorption/Ionization−Time of Flight−Mass Spectrometry) validated the AuNC’s molecular composition, while TEM and DLS measurements confirmed the AuNC’s uniform size and monodispersity [59]. Similarly, anti-Flt1 peptide (AF) AuNCs are synthesized through a one-pot method involving AF and HAuCl_4_ under controlled temperature conditions, forming AuNCs (~1.9 nm) with red fluorescence (~620 nm). AF serves as both a stabilizing ligand and a therapeutic agent, inhibiting angiogenesis by blocking VEGFR1 (vascular endothelial growth factor receptor-1) receptor–ligand interactions [60]. Another promising platform involves Au_25_(S-TPP)_18_ NCs (TPP-SNa = sodium 3-(triphenylphosphonio)propane-1-thiolate bromide) synthesized by reacting Au(tht)Cl (tht = tetrahydrothiophene) and TPP-SNa, followed by NaBH_4_ reduction. The resulting Au_25_(S-TPP)_18_ exhibit a well-defined trigonal crystalline structure, featuring an Au_13_ icosahedral core surrounded by six Au_2_S_3_ motifs, as confirmed by single-crystal X-ray diffraction (XRD). The positive zeta potential (+47.1 mV) from the TPP ligand facilitates mitochondrial targeting by promoting electrostatic interactions with negatively charged mitochondrial membranes. UV-Vis spectroscopy revealed characteristic absorption peaks at 400, 450, and 670 nm, consistent with sulfhydryl-capped AuNC structures. The hydrodynamic size (~2 nm), determined through DLS and TEM, supports efficient renal clearance, falling below the filtration threshold of 5.5 nm. Additionally, the chemical composition of the clusters was confirmed by ESI-MS, showing a mass consistent with theoretical isotopic distributions [61]. Finally, CY-PSMA-1-Au_25_NCs (CY-PSMA-1 = high affinity prostate-specific membrane antigen with additional cysteine and tyrosine residues) were synthesized using a one-step method involving the CY-PSMA-1 peptide and HAuCl_4_, forming highly stable AuNCs with precise prostate-specific membrane antigen (PSMA) targeting due to cysteine and tyrosine residues. DLS and TEM confirmed AuNC sizes (~2 nm), while UV-Vis and fluorescence spectroscopy revealed characteristic absorption and emission peaks. MALDI-TOF-MS verified the expected molecular structure. This comprehensive characterization confirmed the structural integrity and optical properties [62].

### 2.2. Comparison of AuNC Synthesis Methods for Clinical Applications

Comparing synthetic methods for AuNCs is crucial for advancing their clinical application as each method offers unique advantages and limitations related to biocompatibility, scalability, structural precision, and ease of functionalization. Protein-templated synthesis of AuNCs offers a biocompatible and aqueous approach, making it particularly suitable for biomedical applications due to its gentle reaction conditions and compatibility with biological systems. This method produces tunable fluorescent AuNCs with strong photostability, particularly when using larger proteins like BSA. The versatility of this technique stems from its ability to utilize a range of protein templates (e.g., BSA, lysozyme, trypsin), with protein residues such as cysteine playing a critical role in regulating AuNC stability and emission properties. Additionally, protein templates enable multifunctionality and targeting capabilities, making them valuable for various imaging and therapeutic applications. However, the method faces challenges, particularly with smaller proteins like lysozyme and trypsin, which require higher protein-to-Au ratios for effective stabilization. Proteins like pepsin, with low Au-binding residue content, exhibit poor fluorescent performance due to high reduction capabilities that favor the formation of larger non-fluorescent nanoparticles. Protein-templated synthesis is also highly sensitive to environmental factors such as pH, temperature, and UV radiation, which can impact fluorescence stability. Limited cysteine content in some proteins further reduces stability and causes a blue shift in emission wavelengths. Additionally, long-term stability is significantly higher when AuNCs are stored in powder form compared to solutions, presenting a limitation for some applications. Despite these challenges, this method remains a promising approach for the synthesis of functional AuNCs in clinical applications [63,64]. The Brust–Schiffrin method is commonly employed for synthesizing AuNCs due to its versatility in producing size-tunable and highly monodisperse particles through precise ligand and reducing agent control. Its ability to facilitate efficient phase transfer of metal precursors and reducing agents enables high-yield synthesis, making it a valuable method in nanomaterial research. However, its broader use is constrained by challenges such as micelle formation, complex reaction dynamics, and reliance on organic solvents, which pose concerns about safety and biocompatibility [54,65]. Ligand exchange reactions provide a versatile approach to tailoring the surface properties of AuNCs by replacing original ligands with functional counterparts, enabling precise size control, optical tuning, and structural transformation. This method facilitates the formation of thermodynamically stable clusters with improved optical properties like enhanced fluorescence. However, ligand exchange reactions come with challenges, including structural instability due to incomplete ligand exchange, aggregation risk, and batch-to-batch variability. Additionally, precise control of reaction conditions, including ligand type, etching temperature, and reaction time, is crucial as these factors significantly impact the AuNC’s structure, stability, and size distribution [66]. The CO reduction method offers precise control over AuNC size and structure due to its mild reducing power and tunable reaction kinetics. Adjusting the solution pH during synthesis enables the formation of AuNCs with specific optical properties. Its compatibility with various ligands and ability to produce well-defined clusters make it valuable for biomedical applications. However, the method’s reliance on gas-phase CO, slow reaction kinetics, and pH sensitivity poses challenges for scalability. Additionally, specialized gas-handling equipment is required, and yields may be lower compared to conventional reducing agents like NaBH_4_, limiting its large-scale applicability [66]. By understanding the strengths and weaknesses of AuNC synthesis methods, researchers can better tailor their production for clinical applications, enhancing key characteristics crucial for photon therapy such as fluorescence, stability, reduced toxicity, and ensuring improved safety, functionality, and scalability.

## 3. Important Properties of AuNCs

### 3.1. Optical Properties

To target and selectively destroy cancer cells while sparing healthy cells, it is crucial to understand the unique optical properties of Au_25_NCs. These properties can be exploited for cancer diagnosis and treatment. Au_25_NCs can serve as contrast agents in imaging techniques, allow for PTT and PDT to take place, and enhance the efficacy of radiotherapy. The optical properties of Au_25_NCs are essential for developing effective image-guided cancer therapies.

#### 3.1.1. Absorbance

Delving into the absorbance properties of Au_25_NCs is important as these AuNCs exhibit unique absorbance characteristics utilized in PDT and PTT. Aikens et al. observed that the Au_25_NCs display several molecular-like transitions in their absorption spectrum, attributed to strong quantum size effects, with three distinct bands at 670 nm, 450 nm, and 400 nm (Figure 3). In contrast, gold nanoparticles larger than 3 nm generally show surface plasmon resonance at 517 nm. Therefore, the absorption characteristics of Au_25_NCs are primarily influenced by quantum size effects [22]. Furthermore, their (near-infrared) NIR absorbance facilitates deeper tissue penetration, reaching depths of up to ~3 mm with micron-level spatial resolution and a high signal-to-background ratio, making it highly suitable for biomedical imaging and clinical applications [35].

Li et al. explored the impact of different ligands on the absorption characteristics of Au_25_NCs, contrasting those capped with aromatic and aliphatic ligands. Their investigation revealed a red-shift in the UV-Vis spectra of clusters with aromatic ligands, consistent with theoretical predictions based on density functional theory [67]. Yuan et al. investigated the optical properties in bi-thiolate-protected Au_25_NCs when negatively charged thiolate ligands are combined with positively or neutrally charged ones, leading to unexpected absorption peaks at 780 and 980 nm. This surface charge anisotropy induces structural distortion in the NCs, causing abnormal absorptions [68]. In their publication, Tofanelli et al. delve into the intricate details of the Jahn–Teller effect as it manifests in the context of the Au_25_(SR)_18_NC, focusing on the significant consequences of this effect on the Au_25_ absorption spectra. The authors noted that there is a blue shift in the absorption spectrum as the oxidation state increases from −1 to +1, which they attribute to the changes in the electronic structure of Au_25_ caused by the Jahn–Teller effect [69,70].

#### 3.1.2. Emission

Examining the emission properties of Au_25_NCs is important in photon-based cancer therapies as this directly impacts the effectiveness of the therapy. These properties enable their detection, tracking, and precise cancer cell visualization. Fluorescence in AuNCs is influenced by their size, structures, ligands, oxidation states, and geometrical arrangements. Due to quantum confinement, AuNCs display distinct optical traits, showing discrete electronic states and molecule-like behavior. The fluorescence mechanism involves metal-centered free-electron transitions and ligand-to-metal charge transfer (LMCT) processes, influenced by ligand electron donation and density. Geometrical structures and NC core oxidation states also influence optical properties [71]. Tuning these characteristics has emerged as a strategic avenue for achieving bright photoluminescence. Moreover, recent studies have demonstrated enhanced AuNCs with photoluminescence quantum yield (PLQY) values ranging between 40 and 60%, whereas the use of protein templates for AuNCs yields PLQYs around 10% [72]. Weerawardene et al. study the photoluminescence origin of Au_25_(SR)_18_^−^ (R=H, CH_3_, CH_2_CH_3_, CH_2_CH_2_CH_3_) NCs using density functional theory and time-dependent density functional theory calculations. The first excited state causes slight shell expansion, altering electronic structure and inducing substantial Stokes shifts. Multiple excited states, originating from super atom P to D orbital transitions, contribute to emission, with ligands effects primarily through core-based transitions, not charge transfer or semi-ring states [73].

Au_25_NCs possess unique fluorescence properties in the NIR region (shown in Figure 3) that have attracted considerable attention for clinical applications. NIR emission minimizes self-fluorescence and scattering interference, enabling clearer imaging with higher contrast [74]. Dixon et al. examine how BSA-Au(III) complexes fluoresce based on pH, revealing a mixture of blue and red fluorescence at pH around 9.7 due to specific BSA conformation and energy transfers. The gradual emergence of fluorescence, Au(III) incorporation, and molar ratios impacting red fluorescence suggest limits to Au(III) incorporation. The findings challenge single-site nucleation of Au_25_NCs, emphasizing correlations between BSA conformations and fluorescence, with ongoing research to unveil specific Au binding sites and the underlying mechanism [75].

### 3.2. Stability

Stability is a crucial factor in the development of effective cancer therapeutics using Au_25_NCs, as unstable particles can lead to undesired biological effects and poor therapeutic outcomes. Au_25_NCs have been shown to exhibit good stability, making them a promising candidate for cancer therapy. Their increased stability can lead to enhanced circulation time in the bloodstream, increased accumulation at the tumor site, and improved therapeutic efficacy. Therefore, understanding the stability of Au_25_NCs is essential for developing effective and safe cancer therapies.

Au_25_NCs demonstrate remarkable stability under a variety of conditions. Although limited, existing research has examined the thermal stability of Au_25_NCs. According to Teymorian et al., Au_25_NCs synthesized utilizing protein templates demonstrate viable photothermal stability, with no notable changes observed even when exposed to temperatures as high as 80 °C [76]. Investigating the chemical stability of Au_25_NCs is an essential factor to consider when assessing their overall stability. Their chemical stability can be affected by changes in pH, temperature, and exposure to other chemicals and light [77]. Chemical stability can be improved by using biocompatible ligands, optimizing the synthesis conditions and implementing storage and handling methods. Another way to improve chemical stability is by encapsulating the Au_25_NCs in protective matrices, such as silica or polymers [71,78].

Exploring the photostability of Au_25_NCs is a critical aspect to consider in evaluating their overall stability. Jiang et al. evaluated the photochemical stability of Au_25_(SG)_18_. Upon exposure to an 808 nm diode laser with a power of 0.5 W/cm^2^, conventional organic dyes experienced a decrease in NIR absorption ranging from 27% to 72% after 3 min (Figure 4). In contrast, Au_25_(SG)_18_ exhibited nearly unchanged NIR absorption after the same laser exposure. This can be attributed to the superior chemical stability of the Au_25_NC (Figure 4) [40].

Li et al. investigated the photostability of Au_25_NCs in phosphate-buffered saline (PBS), deionized water, and fetal bovine serum (FBS) over 7 days. Au_25_NCs demonstrated stability in each solvent, while FBS further enhanced their fluorescence due to protein interactions. Additionally, the researchers examined Au_25_NCs photostability in different pH solutions, revealing a 1.87-fold increase in NIR-II fluorescence intensity at pH 6.47 compared to pH 7.71. Moreover, under continuous laser radiation, Au_25_NCs in FBS exhibited superior photostability, followed by PBS [30]. An additional study conducted Liu et al. provided insights into the photostability of Au_25_(Capt)_18_NCs through the assessment of four consecutive heating-cooling cycles. Their findings revealed that the temperature increase caused minimal changes, confirming, in addition, thermal stability. Furthermore, singlet oxygen generation was measured using singlet oxygen sensor green (SOSG) at different time points during laser irradiation. After 1 h of continuous irradiation, the production of reactive oxygen species (ROS) continued to increase, indicating a high level of photostability for Au_25_(Capt)_18_NCs [50,79,80].

Stable dispersibility of Au_25_NCs in biological environments is essential for maintaining their therapeutic efficacy and minimizing aggregation-induced toxicity. When functionalized with hydrophilic biocompatible ligands, Au_25_NCs are highly soluble in water, enhancing their dispersibility in aqueous solutions [81]. Wang et al. demonstrated that Au_25_NCs maintained excellent stability in physiological environments, including H_2_O, PBS, FBS, and plasma, showing minimal fluorescence intensity reduction after one day of incubation and stable fluorescence from day 3 to day 7 [82]. Au_25_NCs exhibit exceptional dispersibility in biological systems due to their ultrasmall hydrodynamic size, enabling rapid distribution similar to small molecules. This size facilitates efficient renal clearance, reducing non-specific organ accumulation and minimizing long-term toxicity risk [35]. For example, intravenously administered Au_25_NCs rapidly filtered into the renal pelvis for elimination within approximately 18.5 s, demonstrating swift systemic clearance. This rapid clearance prevents extended circulation, reducing the likelihood of aggregation, instability, and potential toxicity, as discussed further in the next section [40]. Collectively, these properties underscore the potential of Au_25_NCs as stable agents suitable for therapeutic and diagnostic applications in biomedical research.

### 3.3. Toxicity

Understanding the toxicity and long-term safety of Au_25_NCs is essential for their safe and effective utilization in biomedical applications. An in-depth examination of their toxicity profile is crucial to evaluate their biocompatibility, biodistribution, and clearance mechanisms under therapeutic conditions. Liu et al. conducted a comprehensive study on the pharmacokinetics and excretion pattern of Au_25_NCs, highlighting important findings regarding their biosafety [26]. The study revealed that Au_25_NCs, characterized by their small hydrodynamic size, exhibited rapid clearance from the cerebral vessels of mice, resembling the pharmacokinetics of molecules. Whole-body imaging demonstrated the sequential visualization of organs, with the lung, vessel, kidney, and liver becoming discernable at, 1, 3.2, 6.4, and 1.6 s, respectively (Figure 5a–c,g). The kidney was found to be the primary target organ, with gradual intensification of signal over time, peaking at 130 s (Figure 5d–f). Subsequent biodistribution analysis indicated that Au_25_NCs were primarily excreted through urine (86.6%) and feces (2.2%) after 48 h, with very low concentrations observed in the kidney after 7 days. These findings suggest minimal long-term retention in critical organs, supporting favorable pharmacokinetics and reinforcing their potential for biomedical applications [26].

Additionally, Jiang et al. utilized a new imaging technique, combining photoacoustic and ultrasound imaging (using multispectral optoacoustic tomography (MSOT)) to visualize real-time transport of Au_25_NCs in BALB/c mice. This biodistribution study revealed that intravenously administered Au_25_(SG)_18_NCs rapidly filtered into the renal pelvis for elimination, taking approximately 18.5 s to do so. The researchers observed a shorter decay half-life in the renal parenchyma (133.8 s) and the renal pelvis (233.9 s) compared to previously reported GS-AuNPs. Over 60% of the injected dose of Au_25_(SG)_18_ was eliminated within just 30 min post-injection, signifying fast renal clearance, distinguishing it from non-renal clearable gold nanorods [40]. Li et al. conducted in vitro tests on normal NIH/3T3 cells and two types of prostate cancer cells, indicating that Au_25_(SG)_18_ exhibit low toxicity (Figure 6) unless irradiated which induces cytotoxicity. In addition, in vivo tests on C57BL/6 normal mice exhibited no signs of abnormal behavior, and no signs of toxicity were observed. Histological analysis of vital harvested organs from the mice showed no necrotic lesions or edema, indicating high biocompatibility and bio-toleration of Au_25_(SG)_18_ [30].

To further explore the biosafety profile of Au_25_NCs, Wang et al. investigated their pharmacokinetics and toxicity at high doses. Pharmacokinetic studies using NIR-II imaging revealed efficient renal excretion of Au_25_NCs, with 75% clearance achieved within 4 h post-injection and minimal organ retention. Hematological evaluations showed no abnormalities in inflammatory markers or blood cell counts, supporting systemic safety even at high doses (300–500 mg/kg). Histological analysis confirmed the absence of inflammation, necrosis, or tissue damage in key organs such as the liver, kidneys, and brain. Neurotoxicity assessments indicated no signs of neuronal degeneration or cytokine elevation in the brain, suggesting no blood–brain barrier penetration or neuroinflammation. Biomarkers of renal and hepatic function remained within normal ranges, indicating preserved organ function. These results underscore the clinical potential of Au_25_NCs for cancer phototherapy and biomedical imaging, emphasizing their strong biocompatibility, stability, and efficient clearance [82]. The following section shows that unlike larger AuNPs, which clear the body slowly and tend to accumulate in the liver and spleen, potentially causing changes in gene expression and liver necrosis, AuNCs are well below the renal clearance threshold (~5.5 nm). This small size enables efficient urinary excretion, minimizing nonspecific organ uptake by the liver and spleen and reducing the risk of AuNC-induced organ toxicity [62]. Additionally, pharmacokinetic studies involving AuNCs examined their clearance after intravenous administration in CD-1 mice, with doses ranging from 0.15 to 1059 mg/kg. The study observed accelerated renal clearance at doses above 15 mg/kg, indicating a dose-dependent elimination pattern. Additional assessments in nonhuman primates confirmed normal blood pharmacokinetics and the absence of renal abnormalities, supporting the biocompatibility of these AuNCs [83]. While AuNCs are typically regarded as having low toxicity and high biocompatibility, their long-term accumulation and potential effects within the body remain incompletely understood. Further research is essential to evaluate their extended safety profile, particularly given their intended applications in biomedical settings.

Au_25_NCs possess distinct characteristics that render them appealing for various applications, particularly in targeted phototherapy for cancer treatment. Their unique optical properties, including their fluorescence emission in the NIR region, make them valuable for effective treatment. Their stability under various conditions ensures prolonged therapeutic efficacy. Biocompatibility studies demonstrate low toxicity, efficient renal clearance, and minimal organ accumulation, supporting their safe use in biomedical applications. These properties, combined with their customizable surface chemistry, highlight Au_25_NCs as promising candidates for advanced cancer diagnostics and targeted therapies.

## 4. Phototherapy

### Photodynamic Therapy and Photothermal Therapy

Au_25_NCs have demonstrated promising phototherapeutic potential through photosensitization under laser irradiation. Liu et al. showed that Au_25_(Capt)_18_NCs effectively generate singlet oxygen (^1^O_2_) and heat, enabling simultaneous PDT and PTT in cSCC cells and mouse models. Notably, local infiltration of CD4+T and CD8+T cells following treatment highlights potential immune system activation [79]. Katla et al. further confirmed PTT efficacy of Au_25_(SG)_18_NCs against MDA-MB-231 breast cancer cells (Figure 7). PTT was implemented using various laser powers from 6.25 to 10 W/cm^2^ combined with varying irradiation durations (1 to 5 min). Longer irradiation times led to increased cell death, as evident from the fluorescence images in Figure 7a–j. Figure 7k–r show cell viability statistics. Control experiments (Figure 7k–m) demonstrated over 96% cell viability. The combined use of laser and Au_25_(SG)_18_NCs achieved 100% cell death at 10 W/cm^2^ for 5 min. Laser powers of 5 or 6.25 W/cm^2^ had a minimal impact within 5 min, while higher laser powers (7.5–10 W/cm^2^) showed increased impact, reaching 100% cell death at 10 W/cm^2^ for 5 min (Figure 7o–r) [84].

Feng et al. developed (up-conversion nanoparticles) UCNPs@g-C_3_N_4_-Au_25_-PEGNCs (g-C_3_N_4_ = photoactive graphitic-phase carbon nitride, PEG = polyethylene glycol), a multifunctional nanoplatform for NIR laser-mediated image-guided PDT. The platform comprises a mesoporous structure with g-C_3_N_4_ shell coating on UCNPs, ultrasmall Au_25_NCs, and PEG molecules, enabling deep tissue imaging and enhanced PDT efficacy. It exhibited excellent biocompatibility with no toxic effects after 24 h of cell incubation, while (3-(4,5-dimethylthiazol-2-yl)-2,5-diphenyltetrazolium bromide) MTT assays confirmed significant HeLa cell death upon NIR laser irradiation [85]. Jiang et al. reported that GS-Au_25_NCs effectively deliver the FDA-approved NIR dye, indocyanine green (ICG) for cancer PTT while avoiding accumulation in healthy tissues. The resulting ICG_4_-GS-Au_25_NCs demonstrated enhanced photostability, photothermal efficiency, tumor targeting, and prolonged blood circulation, leading to tumor reduction (Figure 8) [48].

Yang et al. developed Fe_3_O_4_/ZIF-8-Au_25_ (IZA) nanospheres, integrating Fe_3_O_4_ nanocrystals for hyperthermic cell elimination and targeted MRI imaging, combined with Au_25_(SR)_18_ NCs for ROS generation under NIR light, achieving synergistic PDT and PTT [86,87,88]. Bi et al. addressed tumor microenvironment challenges by designing a MnO_2_ nanosheet-based nanoplatform integrating Au_25_NCs and platinum (IV) prodrugs, enhancing MRI contrast and enabling chemotherapy-PDT synergy [89]. He et al. synthesized a dual-function platform combining captopril-stabilized Au_25_NCs and Nd^3+^-sensitized UCNPs for multimodal imaging (photoacoustic and up-conversion luminescence) and dual-mode cancer therapy (PTT and PDT) [90]. Fan et al. created Mito-Au_25_@MnO_2_ nanocomposites to target cancer cell mitochondria. This system activates upon GSH-triggered MnO_2_ degradation, inducing oxidative stress and enhancing PDT efficacy [91].

## 5. Radiotherapy

Zhang et al. utilized Au_25_NCs combined with biocompatible coating agents (GSH and BSA) to produce a novel radiosensitizer. The Au core provides strong radiotherapy enhancement, and the coating ligands offer biocompatibility. The GSH-Au_25_NCs resulted in superior cancer radiotherapy enhancement when compared to larger AuNPs. The enhanced radiotherapy effect was attributed to DNA damage caused by the photoelectric effect and Compton scattering of Au_25_NCs. These AuNCs exhibited efficient renal clearance without toxicity (Figure 9) [38].

Luo et al. demonstrated successful targeting of AuNCs to prostate cancer cells that express a specific biomarker. They used CY-PSMA-1 to develop a sensitizer for radiation therapy, showing high PSMA affinity and efficient uptake by tumors with low liver uptake. The targeted NCs improved radiation therapy outcomes in mice, holding promise for further optimization [62]. In another study, Luo et al. increased the efficacy of radiotherapy and chemotherapy by creating a PSMA-AuNC-MMAE conjugate for treating prostate cancer. They functionalized AuNCs with PSMA-1 targeting agents and linked MMAE to AuNC using a cathepsin cleavable bond. The presence of cathepsin in tumor cells triggered the release of active MMAE, demonstrated both in vitro and in vivo. This combination of gold and MMAE synergistically enhanced the effects of radiotherapy, leading to antitumor activity (Figure 10) [92].

Hua et al. designed water-soluble Au_25_(S-TPP)_18_NCs for mitochondria-targeted radio-immunotherapy. Compared to Au_25_(SG)_18_NCs, Au_25_(S-TPP)_18_NCs showed superior radiosensitizing efficiency. This was due to their ability to cause more severe damage to mitochondria upon X-ray exposure, higher production of reactive oxygen species (ROS) with better colocalization in mitochondria, and strong inhibition of intracellular TrxR activity. The enhanced radiosensitizing efficacy of Au_25_(S-TPP)_18_NCs resulted in stronger antitumor immune responses [61]. Fang et al. address the need for improved cancer radiotherapy with reduced side effects by synthesizing mitochondria targeting Au_25_NCs using a peptide template. The CCYKFR-templated Au_25_NCs exhibit enhanced fluorescence, effective targeting of mitochondria, and significant ROS generation under X-ray irradiation, leading to cancer cell death [93].

## 6. Conclusions and Outlook

The therapeutic potential of Au_25_NCs in cancer therapy is considerable, offering enhancement of PDT, PTT, and radiotherapy. Au_25_NCs possess unique attributes that facilitate photon-to-heat conversion and ROS generation, rendering them an ideal candidate for synergistic PDT and PTT or radiotherapy. Au_25_NCs adjustability allows for tailored approaches for specific cancer types. By combining PDT and PTT, Au_25_NCs present a multifaceted strategy for effective cancer therapy, warranting further investigation into optimal conditions and mechanisms. Furthermore, the incorporation of Au_25_NCs in radiotherapy holds the potential to enhance its efficacy while significantly reducing the required dosage by an order of magnitude. Despite these advances, several technical challenges remain that warrant further exploration. A critical bottleneck is the long-term safety profile of Au_25_NCs, particularly regarding potential organ accumulation, chronic toxicity, and immune responses following prolonged exposure. Addressing this requires comprehensive long-term pharmacokinetic and toxicological studies in preclinical models. Furthermore, more studies on biological stability under physiological conditions is crucial for clinical translation. Ultimately, further preclinical studies are necessary to thoroughly evaluate the safety, efficacy, and optimal dosing regimens in various cancer models. This would provide crucial data on the therapeutic effects of Au_25_NCs, potential side effects, and their mechanism of action. Further research is required to better understand the pharmacokinetics, biodistribution, and clearance of Au_25_NCs in the human body. Additionally, scalability and reproducibility in large-scale synthesis also remain significant challenges, as minor batch-to-batch variations can impact Au_25_NC efficacy. Standardizing synthesis protocols and developing scalable, eco-friendly production methods will enhance clinical feasibility.

Based on the reports in this review, it is evident that Au_25_NCs hold potential in cancer therapy. These NCs offer several advantages, such as high biocompatibility, excellent stability, reproducibility, and selective targeting of cancer cells. Additionally, their unique physiochemical properties, including strong luminescence and high photothermal conversion efficiency, enable them to function as excellent theragnostic agents for cancer diagnosis and treatment. The versatility of Au_25_NCs also allows for the delivery of a wide range of anticancer drugs, making them promising candidates for improving the efficacy of chemotherapy. These characteristics make them a promising component for the advancement of biomedical research as they hold the potential to improve the lives of many patients suffering from the side effects of current cancer therapies. With ongoing research, it is evident and important that Au_25_NCs will continue to play an increasingly significant role in the development of novel and effective cancer therapies.

## Figures and Tables

**Figure 1 nanomaterials-15-00039-f001:**
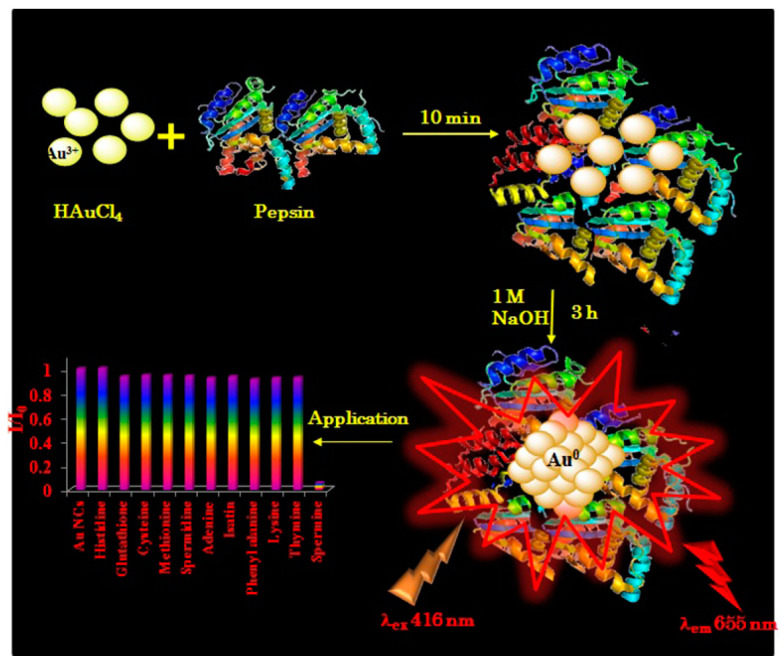
Diagram illustrating a one-step luminescent Au_25_NCs synthesis using pepsin and their application in spermine detection. Reprinted with permission from ref. [52]. Copyright 2019, Elsevier.

**Figure 2 nanomaterials-15-00039-f002:**
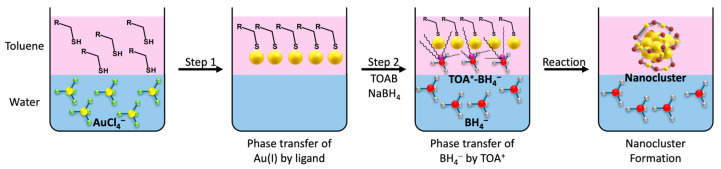
Schematic representation of the Brust–Schiffrin method showcasing the synthesis of AuNCs with a wide variety of ligands.

**Figure 3 nanomaterials-15-00039-f003:**
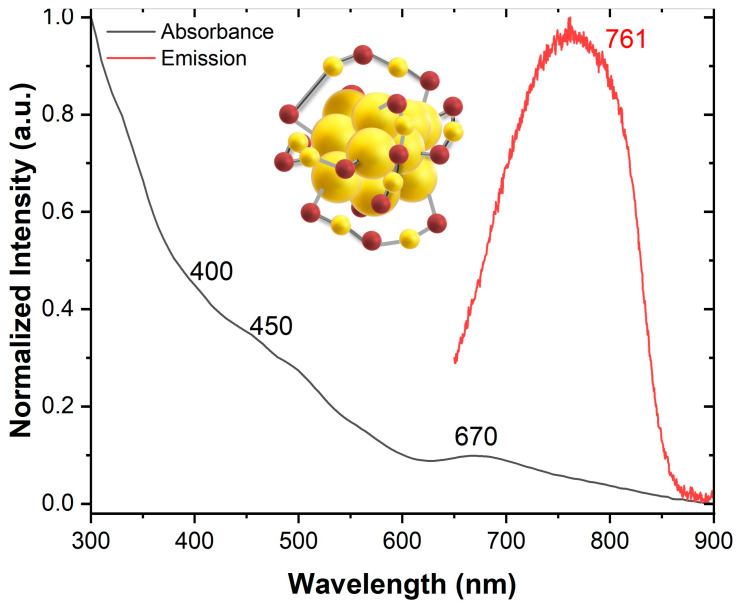
The normalized absorption and emission spectra of Au_25_NCs in water reveal characteristic absorbance peaks at approximately 400 nm, 450 nm, and 670 nm, reflecting their distinct optical properties. The emission peak, observed at 761 nm upon 300 nm excitation, highlights the AuNCs’ fluorescence capabilities. An inset shows the structural model of the Au_25_NC, illustrating its core-shell arrangement with gold atoms (yellow) forming the core and ligand-protected surface atoms (red) stabilizing the structure.

**Figure 4 nanomaterials-15-00039-f004:**
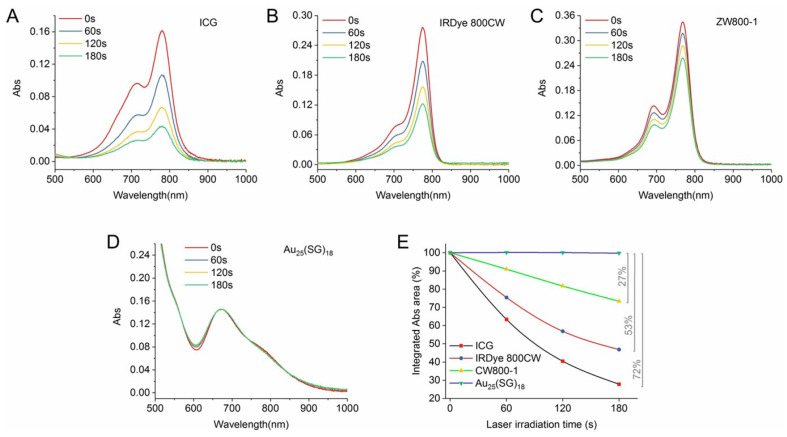
Absorption characteristics of (**A**) ICG, (**B**) IRDye 800, (**C**) ZW800-1, and (**D**) Au_25_(SG)_18_ were examined with a 3 min exposure to an 808 nm diode laser (0.5 W/cm^2^). (**E**) NIR absorption changes (659–950 nm) during the 3 min irradiation were tracked. Reprinted with permission from ref. [40]. Copyright 2019, John Wiley and Sons.

**Figure 5 nanomaterials-15-00039-f005:**
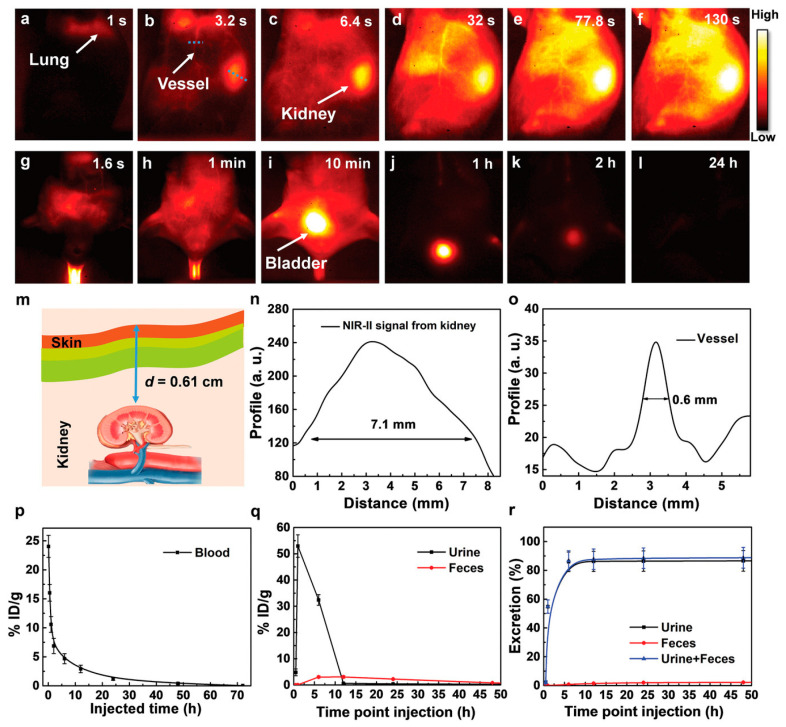
(**a**–**l**) Rapid NIR-II fluorescence imaging (1300 nm filter, 300 ms exposure) of whole-body mouse after AuNCs infusion, showing rapid excretion within 24 h. Excitation at 140 mW/cm^2^, 808 nm. (**m**) Schematic of kidney penetration. (**n**–**o**) Mouse kidney and vessel description for image (**b**). (**p**) Short blood half-life of AuNCs via ICP-MS. (**q**) ICP-MS tracking of evolving AuNCs concentrations in urine and feces. (**r**) Exceptional 4 -h cumulative urine, feces, and combined excretion of AuNCs. Reprinted with permission from ref. [26]. Copyright 2019, John Wiley and Sons.

**Figure 6 nanomaterials-15-00039-f006:**
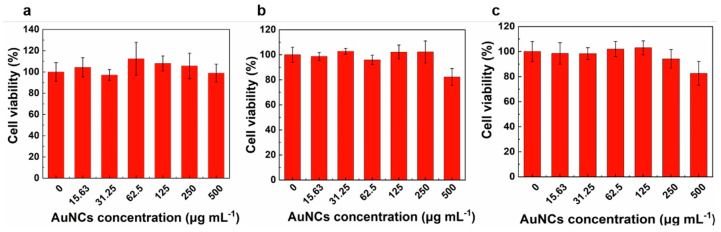
AuNCs showed no cytotoxicity toward normal NIH/3T3 cells, two prostate cancer cell lines, or Trop2-overexpressing DU145-Trop2-OV cells. Cell toxicity assessed using 3-(4,5-dimethylthiazol-2-yl)-2,5-diphenyltetrazolium bromide (MTT) assay for (**a**) mouse fibroblast NIH/3T3 cells; (**b**) human prostate cancer DU145 cells; (**c**) DU145-Trop2-OV cells with heightened tumor growth and metastasis due to Trop2 overexpression. All experiments were triplicated, with error bars representing standard deviation. Reprinted with permission from ref. [30]. Copyright 2020, John Wiley and Sons.

**Figure 7 nanomaterials-15-00039-f007:**
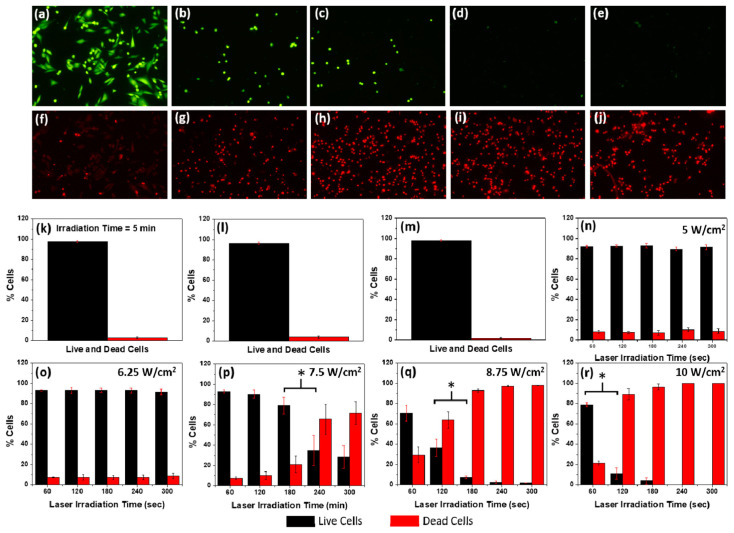
Photothermal therapy assessment with Au_25_(SG)_18_NCs on MDA-MB-231 cells, stained with (**a**–**e**) calcein-AM and (**f**–**j**) EthD-1 after 1–5 min of laser exposure (10 W/cm^2^, 1-min intervals). Cell viability histograms for various conditions: (**k**) laser-only (10 W/cm^2^, 5 min); (**l**) 0.75 mg/mL NCs, no laser; (**m**) no NCs, no laser; (**n**–**r**) 0.75 mg/mL NCs, laser exposure at 5, 6.25, 7.5, 8.75, and 10 W/cm^2^, respectively, for 1–5 min. Student’s *t*-test analyzed data, with * indicating significance in (**p**–**r**) at *p* < 0.05. Reprinted with permission from ref. [84]. Copyright 2018 American Chemical Society.

**Figure 8 nanomaterials-15-00039-f008:**
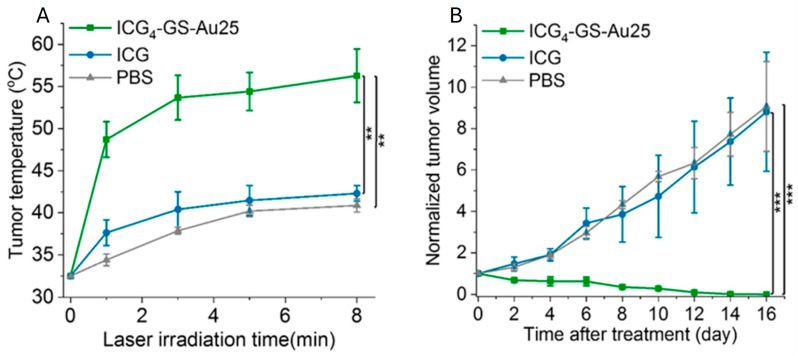
(**A**) Mice received ICG_4_-GS-Au_25_, free ICG, or PBS intravenously, and tumor temperatures were monitored under 0.8 W/cm^2^ 808 nm laser (8 min, *n* = 3/group). (**B**) Tumor volumes post-PTT were analyzed (two-sample *t*-test, ** *p* < 0.005, *** *p* < 0.0005). Reprinted (adapted) with permission from [48]. Copyright 2020. American Chemical Society.

**Figure 9 nanomaterials-15-00039-f009:**
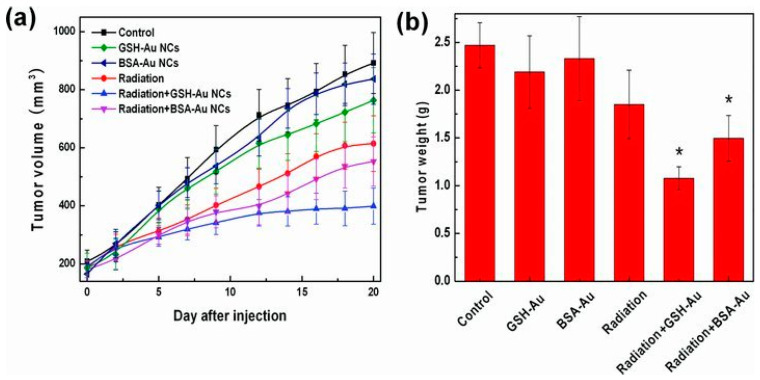
Examined were tumor (**a**) size changes and (**b**) mass variations in mice treated with GSH- and BSA-Au_25_NCs (10 mg-Au kg-body^−1^). Data were evaluated via Student’s *t*-test, with * indicating *p* < 0.05 in (**b**). Reprinted with permission from ref. [38]. Copyright 2013, John Wiley and Sons.

**Figure 10 nanomaterials-15-00039-f010:**
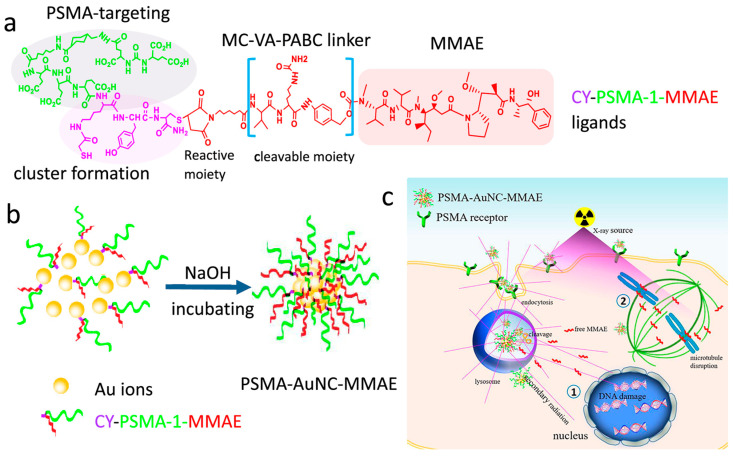
(**a**) CY-PSMA-1 MMAE ligand molecular structure; (**b**) formation of PSMA-AuNC-MMAE conjugates; and (**c**) Internalization process of PSMA-AuNC-MMAE conjugates, followed by intracellular release of MMAE from lysosomes, enhanced radiosensitivity due to gold (1), and chemotherapy effects through the microtubule disruption mechanism of free MMAE (2). Reprinted (adapted) with permission from [92]. Copyright 2022, American Chemical Society.

**Table 1 nanomaterials-15-00039-t001:** Comparison of gold nanostructure properties across different sizes.

Particle Size	Synthesis Degree of Difficulty	Reproducibility	Biocompatibility	Cellular Uptake	Tumor Uptake	Therapeutic Efficacy Upon Irradiation	Elimination of Particles
Au_25_NCs	+ [19]	+ [25,26]	+ [30,31]	+ [34,35]	+ [38]	+ [38,39]	+ [26,31,40]
2–5 nm	+ [23]	− [23]	− [32]	− [15]	+ [15]	− [41]	+ [15]
10–30 nm	+ [23,24]	− [27,28,29]	− [27,42]	+ [15,24,43]	− [15]	− [41]	− [42]
50–100 nm	+ [24]	− [28]	− [33]	+ [28,44,45]	− [46]	− [41]	− [42]

+ Enhanced Ability; − Unable or less proficient.

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
