# Peer review of "Applications of Au25 Nanoclusters in Photon-Based Cancer Therapies"

_nanomaterials, 2024, doi:10.3390/nano15010039_

Round 1

Reviewer 1 Report

Comments and Suggestions for Authors

The manuscript reviews the potential of gold nanoclusters (Au NCs) in photon-mediated cancer therapy, covering areas such as photodynamic therapy (PDT), photothermal therapy (PTT), and radiotherapy (RT). The topic of this article is of great significance, especially in the rapid development of gold nanoclusters in cancer therapy. Although the content is detailed, there is still room for improvement in logical clarity, depth, and literature citation standards.

1. The review summarizes the unique properties of Au NCs and their advantages in cancer treatment, especially in enhanced phototherapy and immunotherapy. This provides a wide range of reference value for researchers. However, the discussion of some areas is still insufficient. For example, a comparison of the advantages and disadvantages of Au NCs synthesis methods can be added, especially in the context of clinical translation. In addition, the discussion of toxicity and long-term safety can be more detailed.

2. It is recommended to further condense the phototherapy section to highlight the key findings. The conclusion section clearly summarizes the prospects of Au NCs in cancer treatment, but lacks specific suggestions for future research directions. It is recommended to supplement the analysis of technical bottlenecks and possible solutions.

4. The references to the figures and tables are clear and supportive, but the layout of the figures should be optimized so that the font size is as consistent as possible for easy reading.

5. The location of some references (such as reference 16) is not clearly marked in the text. Careful proofreading is recommended.

Reviewer 2 Report

Comments and Suggestions for Authors

This manuscript describes the application of Au25 nanoclusters for cancer photo-therapy. I believe this topic is hot and attractive. The authors broadly introduce Au25NCs and their applications for cancer photo-therapy. As mentioned above, this topic is important, and many reviews have been reported. I didn't find a significant difference from others and critical focus of this review. In this review, the authors focus on Au25NCs. If this is the focus, the authors should compare their characteristics with other nanoparticles more precisely. The authors may also summarize more recent advances on this topic. Anyway, the authors should clarify their focus on this review in the introduction and emphasize those points in each section. Thus, I believe this manuscript needs significant revision for publication. 

Major comments

1. This review only focused on the Au25 cluster. This should be clarified in the title, too.  

2. In the abstract, the authors mentioned, "Their small size enhances cellular uptake, prolongs circulation, and reduces liver toxicity compared to larger gold nanoparticles." However, there is no related explanation for cellular uptake, except showing Table 1, and for prolonged circulation. Also, the authors should explain more carefully for Table 1 to clarify their advantages to AuNPs with larger sizes.   

3. The authors explain details of examples for photo-thermal therapy using Au25(SG)18 in the introduction with Figure 1. However, this should be described in the following section, not in the introduction. 

4. Section 2 is "Synthesis & characterization of targeted AuNCs for cancer therapy". The authors describe the synthesis of AuNCs. But I don't see "targeted AuNCs for cancer therapy" here. Fig. 3 shows the synthetic approach, but AuNCs were prepared in organic solvent (toluene). Are these "targeted AuNCs for cancer therapy"?

The authors clearly describe surface properties related to surface modifiers. 

5. Section 3.1 "optical properties" are described as important properties of AuNCs.

The Authors listed 3.1.1 absorbance and 3.1.2 Emission. They showed normalized absorption spectra in Fig. 4, but not emission spectra. They should show it and explain how effective this is. That is related to in vivo imaging, such as shown in Fig. 6a-l.

6. Section 3.2 "stability" is also an important parameter. The authors mention about chemical stability and photo-stability. However, for biological applications, dispersibility is also an important factor related to stability. Thus, they should also describe stable dispersibility in biological conditions.

7. In section 3.3 "toxicity", the authors concluded, "These studies demonstrate their strong NIR absorbance". But there is no data or discussion. I can understand there is NIR absorption due to their fluorescence imaging result. However, is this really strong absorption? Absorption spectra of Au25NCs are shown in Fig. 4 and Fig. 5D. Based on these spectra, how they can say "strong NIR absorbance" compared with what?  

Minor comments

1. The authors mentioned glutathione is a protein in this manuscript several times. But glutathione is not a protein but just a peptide. Polypeptides are proteins, but not all peptides are proteins. 

2. Abbreviations need to be clarified their original names, such as "PLQY", "UCNP", and "g-C3N4" at their first appearance.

Reviewer 3 Report

Comments and Suggestions for Authors

The review by Lockwood et al. concerns a very interesting, promising and currently underexplored nanomaterial for application in cancer phototherapies.

Overall, the manuscript is well structured, however, I have a few concerns/suggestions that should be addressed before publication:

- Figure 2 is not fully readable (red text). Please increase the resolution.

- The sentence (lines 113-115) “Absence of larger Au nanoparticles in the visible range (400-800 nm)” should be written more appropriately. I believe the authors are referring to the absence of plasmon resonance in the range 400-800 nm, which is a characteristic property of larger Au nanoparticles.

- The nanoclusters synthesis part (Section 2) should be improved and the synthetic protocols discussed more clearly. Additionally, Figure 3 should be cited in the text where it is discussed.

- Section 4 "Phototherapy" should be expanded.

- Figure 9c is not fully readable. Please increase the resolution.

Finally, a general doubt concerns the special issue for which the manuscript is intended; in my opinion, a special issue focused on biomedicine would be more appropriate.

Round 2

Reviewer 2 Report

Comments and Suggestions for Authors

The authors significantly revised their manuscript based on the reviewers' comments, making it much better than the previous one. However, some points still need to be revised. Thus, I recommend that this manuscript be published after further revision.

Major comments

1. Prolonged blood circulation is one of the important issues for drug delivery. In some parts, the authors mentioned, "their increased stability can lead to enhanced circulation time in the body, increased accumulation at the tumor site, and improved therapeutic efficacy" (line 346), and "prolonged blood circulation leads to tumor reduction" (line 521). On the other hand, "Their ultrasmall size allows for rapid renal clearance, minimizing the risk of prolonged accumulation in non-target tissues." (line 76), "Their rapid renal clearance ensures minimal long-term retention in the body, reducing the likelihood of chronic toxicity and addressing a common concern associated with larger nanoparticles.26, 31, 40" (line 90).  

I understand that adequate blood circulation and significant clearance mechanisms are both important. However, this manuscript describes these points separately and seems contradictory, which could mislead readers. The authors should clearly explain these points.   

2. Table 1 (comparison of Au25NCs and gold nanoparticles) shows the difference in cytotoxicity between Au25NCs and other AuNPs over 2 nm in diameter. The reason should be described in the introduction (line 79). Is that really a size effect rather than surface ligands?  

Minor comments

1. In this manuscript, "targeted" often comes out. Some of them, such as "targeted phototherapy" and "targeted cancer treatments" are fine. But others, such as "targeted AuNPs" line 52 and line 182 etc. are unclear or improper. 

2. "Their ultrasmall size allows for rapid renal clearance, minimizing the risk of prolonged accumulation in non-target tissues." (line 76)

"Their rapid renal clearance ensures minimal long-term retention in the body, reducing the likelihood of chronic toxicity and addressing a common concern associated with larger nanoparticles.26, 31, 40" (line 90) 

These sentence are in the same paragraph. These seem duplicated.  

3. Table 1 (comparison of Au25NCs and gold nanoparticles) includes "synthesis degree of difficulty" "reproducibility", "toxicity", "cellular uptake", "tumor uptake", "efficacy of upon irradiation", and "elimination". "efficacy of upon irradiation" and "elimination" are insufficent description. Efficacy of what? Elimination of what or from where? should be explained. Also, Table 1 shows + and -. "+" means "enhanced ability". In the case of toxicity, Au25NCs showed enhanced ability of toxicity. This may cause misleading. 

Reviewer 3 Report

Comments and Suggestions for Authors

The authors have improved the quality of the manuscript significantly. I recommend the publication in its present form.

Round 3

Reviewer 2 Report

Comments and Suggestions for Authors

The authors have revised their manuscript well. Now, I can recommend this manuscipt for publication.